# Relative Contribution of Blood Pressure and Renal Sympathetic Nerve Activity to Proximal Tubular Sodium Reabsorption via NHE3 Activity

**DOI:** 10.3390/ijms24010349

**Published:** 2022-12-26

**Authors:** Roberto B. Pontes, Erika E. Nishi, Renato O. Crajoinas, Maycon I. O. Milanez, Adriana C. C. Girardi, Ruy R Campos, Cassia T Bergamaschi

**Affiliations:** 1Cardiovascular Division, Department of Physiology, Escola Paulista de Medicina, Universidade Federal de São Paulo, São Paulo 04021-001, Brazil; 2Heart Institute (InCor), University of São Paulo Medical School, São Paulo 01246-903, Brazil

**Keywords:** NHE3, glomerular filtration rate, bicuculline, paraventricular nucleus of the hypothalamus, sympathetic nerve activity

## Abstract

We examined the effects of an acute increase in blood pressure (BP) and renal sympathetic nerve activity (rSNA) induced by bicuculline (Bic) injection in the paraventricular nucleus of hypothalamus (PVN) or the effects of a selective increase in rSNA induced by renal nerve stimulation (RNS) on the renal excretion of sodium and water and its effect on sodium-hydrogen exchanger 3 (NHE3) activity. Uninephrectomized anesthetized male Wistar rats were divided into three groups: (1) Sham; (2) Bic PVN: (3) RNS + Bic injection into the PVN. BP and rSNA were recorded, and urine was collected prior and after the interventions in all groups. RNS decreased sodium (58%) and water excretion (53%) independently of BP changes (*p* < 0.05). However, after Bic injection in the PVN during RNS stimulation, the BP and rSNA increased by 30% and 60% (*p* < 0.05), respectively, diuresis (5-fold) and natriuresis (2.3-fold) were increased (*p* < 0.05), and NHE3 activity was significantly reduced, independently of glomerular filtration rate changes. Thus, an acute increase in the BP overcomes RNS, leading to diuresis, natriuresis, and NHE3 activity inhibition.

## 1. Introduction

The sympathetic nervous system (SNS) controls the sodium and water balance by the kidneys and blood pressure (BP). The SNS influences renal function by inducing renal vasoconstriction, stimulating renin secretion, and increasing tubular sodium reabsorption [1]. Sympathetic hyperactivity results in excessive vasoconstriction and hypertension [2]. During acute SNS activation, both arterial hypertension and sympathetic actions in the kidneys control sodium and water excretion. However, the relative contribution to salt and water balance by the kidneys of each of these mechanisms is not completely understood.

The brain controls SNS activity through specific nuclei, such as the paraventricular nucleus of the hypothalamus (PVN). Experimentally inhibiting PVN decreases the BP and sympathetic vasomotor tone [3], whereas activation of PVN has the opposite effects [4]. Cardiovascular actions can be evaluated by targeting this area with the GABAa receptor antagonist bicuculline (Bic), which blocks the inhibitory synaptic influence on this crucial cardiovascular control area [5].

Renal nerve stimulation (RNS) is a technique used to examine the specific influence of SNS on renal function without altering the BP. For instance, low-frequency (LF) RNS causes sodium and water reabsorption at the tubular level without affecting renal blood flow and the glomerular filtration rate [6]. Additionally, Healey et al. showed that LF-RNS results in sodium retention, primarily by increasing proximal tubule sodium reabsorption, which is mediated by sodium hydrogen exchanger isoform 3 (NHE3) [7]. Because RNS can be applied at different frequencies, this technique enabled us to demonstrate that SNS activation in the kidney can impact renal tubular function independently of hemodynamic changes [8]. We found that ability of NHE3 to increase LF-RNS in the renal proximal tubule was accompanied by activation of intrarenal but not circulating angiotensin II (Ang II); this effect was completely abolished when the animals were pretreated with the Ang II type 1 receptor antagonist losartan [8] and occurred independently of BP changes_._ Therefore, NHE3 hyperactivation in response to RNS depends on Ang II-induced Ang II type 1 receptor activation in the kidneys [8,9].

NHE3 is the renal apical transporter responsible for most sodium and water reabsorption at the proximal tubular level [10,11]. Thus, NHE3 plays an essential role in maintaining the extracellular water balance, and natriuresis and diuresis are physiologically required under increased BP to eliminate extracellular fluid. The primary role of NHE3 in pressure natriuresis was recently demonstrated by Li et al. using proximal tubule-specific NHE3 knockout mice (PT-Nhe3^−/−^). The authors demonstrated that Ang II-induced hypertension was attenuated in PT-Nhe3^−/−^ and that the mice displayed more prominent pressure natriuresis in response to increased renal perfusion pressure [12].

Increased renal sympathetic nerve activity (rSNA) activates NHE3-mediated sodium reabsorption [7,8], whereas an increased BP reduces this effect [11,13]. However, how proximal tubule NHE3-mediated sodium reabsorption is regulated in the presence of both rSNA stimulation and increased BP remains unclear. Importantly, sympathetic vasomotor activation and arterial hypertension occur in different experimental hypertension models [14,15,16,17] and hypertensive patients [18,19,20]. The PVN plays a major role in central cardiovascular and volume control [2]. Therefore, we target the PVN and examined the effect of acute increases in BP and rSNA induced by Bic injection or the effects of a selective increase in rSNA by RNS on the renal excretion of sodium and water and its potential effect on the regulation of NHE3 activity in uninephrectomized Wistar rats.

## 2. Results

### 2.1. First and Second Series of Experiments

#### 2.1.1. Bic Injection into the PVN Increases BP and rSNA

We evaluated the effect of Bic injection into the PVN on the BP and rSNA. The BP and rSNA were recorded in the sham group, in this group there was no injection of bicuculline in the PVN. The Bic PVN group was injected with Bic into the PVN at baseline, which led to an increase in both rSNA and BP.

Figure 1A shows representative tracers of the pulsatile BP and rSNA in the sham group (throughout the experiment) and Bic PVN group (at baseline before Bic infusion). Figure 1B shows the tracers after Bic infusion. The sham and Bic PVN groups before Bic infusion showed similar values for the mean arterial pressure (MAP) and rSNA. After Bic PVN infusion, the MAP and rSNA increased by ~30% (Figure 1C) and ~60% (Figure 1D), respectively (MAP: sham 121 ± 6 mmHg, pre-Bic PVN 123 ± 5 mmHg, post-Bic PVN 160 ± 5 mmHg; and rSNA: sham 100 ± 3, pre-Bic PVN 95 ± 3, post-Bic PVN 159 ± 7%).

#### 2.1.2. Bic Injection in the PVN Increases Urinary Flow Rate and Urinary Sodium Rate but Not Creatinine Clearance

The effect of acute increases in the BP and rSNA on renal excretory function was evaluated as creatinine clearance (CrCl, to estimate the glomerular filtration rate), urinary flow rate (UFR), and urinary sodium rate (UNa). Soon after the BP and rSNA increased, both UFR and UNa increased significantly, with no change in the CrCl.

The CrCl values of rats in the sham and Bic PVN groups before and after Bic infusion were similar (0.0018 ± 0.0002, 0.0018 ± 0.0002, and 0.0018 ± 0.0001 mL/min/g, respectively; Figure 2A). UFR and UNa showed similar values in the sham and Bic PVN groups before Bic infusion. After Bic infusion, the UFR was 5-fold higher than that at baseline (0.0349 ± 0.0114, 0.0300 ± 0.0037, and 0.1500 ± 0.0182 mL/min/g in the sham, pre-Bic PVN, and post-Bic PVN groups, respectively; Figure 2B), and UNa was increased by ~2.3-fold (1.72 ± 0.06, 1.71 ± 0.07, and 3.95 ± 0.16 µEq/min/g in the sham, pre-Bic PVN, and post-Bic PVN groups, respectively), Figure 2C.

#### 2.1.3. Bic Injection in the PVN Increases NHE3 Phosphorylation Levels at Serine 552

To investigate the mechanism of diuretic and natriuretic pressure in more detail, we evaluated whether these effects were associated with NHE3 inhibition. We assessed the NHE3 phosphorylation levels at serine 552, a surrogate for NHE3 inactivation [21,22,23,24]. Figure 3A shows representative immunoblotting and semi-quantitative results for the NHE3 total abundance, NHE3 phosphorylated at serine 552 (NHE3-PS552), and actin in the renal cortex of sham and Bic PVN rats. Total NHE3 expression did not differ between groups (sham: 100 ± 10% and Bic PVN: 122 ± 15%, Figure 3B), whereas the ratio of NHE3-PS552 to total NHE3 was 49% higher in the Bic PVN group than in the sham group (Figure 3C).

### 2.2. Third Series of Experiments

#### 2.2.1. Effect of RNS and RNS + Bic PVN on BP

To evaluate the diuretic and natriuretic pressure under conditions known to decrease urinary excretion, we used a separate group of anesthetized animals. After baseline BP and rSNA recordings, the animals were first subjected to RNS, which did not affect the BP. Under RNS, animals were injected with Bic into the PVN, following which the BP increased.

The bars in Figure 4 represent the MAP values obtained from the same rats and from the experiment at three different time points. The MAP values were similar at baseline and under RNS. Under RNS, Bic PVN increased the MAP by approximately 33% (122 ± 3 mmHg vs. 125 ± 3 mmHg vs. 163 ± 5 mmHg, respectively).

#### 2.2.2. Effect of RNS and RNS + Bic PVN on Renal Function: CrCl, UFR, and UNa

To investigate the role of RNS alone and with a concomitant increase in BP caused by Bic, we evaluated the CrCl, UFR, and Una at three different periodstime points (baseline, RNS and RNS + Bic). RNS decreased both the UFR and UNa. Under RNS after Bic PVN, the increase in BP was followed by diuresis and natriuresis. The CrCl remained unchanged throughout the experiment.

The bars in Figure 5A represent the CrCl at three different time points in the experiment (baseline: 0.001943 ± 0.0002 mL/min/g, RNS: 0.002029 ± 0.0001 mL/min/g, and RNS + Bic PVN: 0.002072 ± 0.0001 mL/min/g). Compared to the value at baseline, the UFR was significantly decreased by RNS and increased by Bic PVN (0.0358 ± 0.0012 vs. 0.0168 ± 0.0008 vs. 0.0594 ± 0.0020 mL/min/g, respectively) (Figure 5B). Compared to the value at baseline, the UNa was significantly decreased by RNS and increased by Bic PVN (1.54 ± 0.05 vs. 0.64 ± 0.05 vs. 4.06 ± 0.11 µEq/min/g, respectively) (Figure 5C).

## 3. Discussion

We found that bicuculline injection in the PVN increased the rSNA and MAP; the acute increase in BP was followed by diuresis and natriuresis, whereas CrCl remained unchanged. In rats administered Bic in the PVN, NHE3 showed increased phosphorylation at serine 552. Additionally, RNS decreased the UFR and UNa independently of changes in the BP. Under these conditions, an acute increase in the BP (by Bic into the PVN) reverted the effects of RNS and resulted in diuresis and natriuresis.

Disinhibition of the PVN, a vital nucleus containing pre-motor sympathetic neurons, results in a pressor response. The acute increase in BP resulted in substantial pressure diuresis and natriuresis, accompanied by increased NHE3 phosphorylation at serine 552. We showed that in a model of acute increase in centrally generated BP, NHE3 phosphorylation is necessary to counteract the sympathetic antidiuretic and anti-natriuretic effects. We injected Bic into the PVN to increase the BP rather than using vasoactive drugs such as phenylephrine to avoid a reflex decrease in rSNA via the baroreceptor reflex [25,26]. As mentioned previously, PVN inhibition decreases the BP [3], whereas PVN activation increases the BP [4], which is sympathetically mediated. There are two different groups of PVN neurons: magnocellular and parvocellular. Porter and Brody stimulated one of two different groups of PVN neurons and observed that magnocellular stimulation led to vasodilation, whereas parvocellular stimulation resulted in vasoconstriction [27]. In this study, we targeted parvocellular neurons because they project into the medulla and spinal cord, which are also involved in autonomic BP regulation [2,28]. Based on our histological images combined with autonomic and pressure responses, we predominantly activated parvocellular neurons.

We found that the acute centrally generated increased BP was rapidly followed by diuresis and natriuresis associated with higher renal cortical NHE3-PS552 levels. This effect was achieved despite the increase in rSNA, a well-known stimulus of sodium and water retention, without a change in BP [1,6]. Importantly, a correlation between NHE3-PS552 and proximal tubule NHE3 activity has been demonstrated in several experimental models, including in hypertensive animals [8,22,23,24,29,30,31]. Indeed, adult hypertensive SHRs exhibit higher NHE3 activity, which correlates with higher renal cortical NHE3-PS552 [22,23]. Another study by the same group showed that a vitamin D-free diet for 30 days led to hypertension in Wistar rats. Molecular experiments revealed higher NHE3-PS552 levels in the renal cortex and medulla of vitamin D-deficient rats than in control rats, which may be necessary to counteract the increased sodium reabsorption mediated by other renal sodium apical transporters [32]. McDonough’s laboratory also focused on determining the role of the proximal renal tubule in response to pressure natriuresis. In Ang II-dependent hypertensive rats, increased diuresis and natriuresis were accompanied by higher NHE3-PS552 [13]. In another study, female rats excreted sodium more rapidly than did male rats in response to saline challenge, and female rats displayed higher levels of NHE3-PS552 compared to males [24]. Our findings complement those of earlier studies and support that NHE3 phosphorylation at serine 552 is an essential mechanism underlying the pressure natriuresis response.

We assessed the role of RSN alone and its association with increased BP in renal excretion. RNS is a widely used method that mimics increased rSNA and can be used to determine the role of the SNS in renal function [1,6]. Many studies have demonstrated that RNS increases sodium and water tubular reabsorption independently of glomerular or hemodynamic changes [1,33]. We previously showed that RNS results in antidiuresis and anti-natriuresis and is associated with a reduction in NHE3-PS552. Based on these previous findings and the observation that Bic PVN produced pressor diuresis and natriuresis associated with higher renal cortical NHE3-PS552 levels, we assessed a separate group of rats under three different conditions to distinguish between BP and sRNA regulation of sodium and water reabsorption. First, the BP of the rats was recorded for 1 h, following which the renal nerve was stimulated using a frequency that causes tubular sodium and water retention [8,9]. Finally, under RNS, Bic was injected into the PVN to cause an immediate increase in BP. Urine samples were collected at this time point. RNS increased sodium and water reabsorption compared to that at the baseline period, as previously demonstrated [8]. Furthermore, Bic in the PVN produced renal sodium and fluid excretion values greater than those of the two previous periods in the same rats. Although RNS is a potent stimulus of renal sodium and water retention, the increase in BP is a more powerful stimulus for controlling extracellular fluid. However, we did not examine the role of the renal nerve opposing natriuresis and diuresis under conditions such as chronic hypertension, in which both rSNA and BP are chronically increased. A protocol using renal denervation may clarify this mechanism.

### 3.1. Limitations of the Present Study

The experiments were carried out in anesthetized rats, and it is not possible to state that the same findings will be obtained in awake rats. Assessing the PVN and recording sympathetic activity simultaneously in non-anesthetized rats is a difficult task and only new experiments will be able to clarify this question. However, to overcome this limitation, the rats used in the three independent series of experiments were submitted to the same experimental conditions, including the anesthesia protocol.

### 3.2. Conclusions

The major new finding of the present study is that an acute increase in the BP induced by Bic in the PVN overcomes RNS, leading to diuresis, natriuresis, and increase in the NHE3 phosphorylation levels at serine 552, Figure 6 schematically shows this hypothesis. NHE3-mediated sodium reabsorption by renal sympathetic activation may be a new therapeutic target to treat hypertension.

## 4. Materials and Methods

### 4.1. Materials

Ketamine and xylazine were obtained from Syntec (Cotia, Brazil). Sodium thiopental was obtained from Cristália (Itapira, Brazil). The monoclonal antibody to NHE3 was a gift from Dr. Peter Aronson (Yale University, New Haven, CT, USA). A phosphor-specific monoclonal antibody that recognizes NHE3 only when it is phosphorylated at serine 552 [21] was purchased from Santa Cruz Biotechnology (Dallas, TX, USA), and a monoclonal antibody to actin (clone JLA20) was purchased from Merck (Darmstadt, Germany). Horseradish peroxidase-conjugated goat anti-mouse and goat anti-rabbit secondary antibodies were purchased from Life Technologies (Carlsbad, CA, USA). All other reagents were purchased from Sigma Chemical (St. Louis, MO, USA) unless otherwise stated.

### 4.2. Animals

All experimental procedures were performed according to the guidelines established by the Institutional Animal Care Committee of the Ethics in Research Committee of the Universidade Federal de São Paulo (protocol no. 0361/11). Male Wistar rats (*n* = 19, 250–300 g, 8–9 weeks old) were obtained from the animal care facility at the Universidade Federal de São Paulo. All animals were housed in group cages, provided free access to rat chow and water, and maintained in a temperature-controlled environment (23 °C) on a 12:12 h light-dark cycle. The animals were subjected to right nephrectomy, as described previously [8], to avoid any influence from the contralateral kidney. The animals were administered meloxicam (0.1 mg/kg) on the day of surgery and on the two following days.

### 4.3. Study Design

The rats were divided into three groups for the experiments: (1) sham, in which the BP and rSNA were recorded, and urine was collected over 2 h; (2) Bic PVN, in which the BP and rSNA of the rats were recorded, and urine was collected for 1 h at baseline conditions and during 1 h after the GABA-A antagonist Bic was infused into the PVN; and (3) RNS + bicuculine injection into the PVN, after which the BP and rSNA were recorded, and urine was collected for 1 h after RNS and 1 h after Bic injection into the PVN.

In three series of experiments 7–8 days after uninephrectomy (right kidney) to avoid possible effects from the contralateral kidney by the reno-renal reflex [1], the rats were anesthetized (sodium thiopental; 60 mg/kg intraperitoneally) and tracheotomized. The femoral vein was catheterized with a PE-10 polyethylene tube connected to a PE-50 for administration of additional anesthetic (10 mg (kg·h) intravenously) and isotonic saline at 10 mL/(kg·h) in the three independent series of experiments. The femoral artery was cannulated to directly measure the arterial pressure. The temperature of the rats was maintained at 37 °C using a rectal probe connected to a servo-controlled electric blanket (Letica^®^, Woonsocket, RI, USA). The femoral artery catheter was connected to a pressure transducer, and the arterial pressure was recorded online (PowerLab, ADInstruments, Sydney, Australia). The ureter was catheterized with polyethylene tubing (PE-10) to collect the urine.

### 4.4. rSNA Signal Acquisition

A retroperitoneal incision was made to expose the left kidney and record rSNA. Using a dissecting microscope and fine forceps, the renal nerve was localized, freed from the connective tissue, and positioned on a bipolar silver recording electrode. The renal nerve and electrode were covered with paraffin oil. The signal of the renal nerve was displayed on an oscilloscope, and nerve activity was amplified (gain 20,000, Neurolog, Digitimer, Hertfordshire, UK), filtered with a band-pass filter (100–1000 Hz), and collected for display and analysis using a PowerLab data-acquisition system (ADInstruments).

In the third series of experiments (RNS + BicPVN), as previously described [8,9], after BP was recorded for 1 h (baseline), we applied RNS for 1 h using parameters known to cause sodium and water retention without altering the renal blood flow and glomerular filtration rate [6]. Furthermore, for an additional 1 h, the RNS was maintained, and Bic was injected into the PVN to increase the BP (RNS + Bic PVN). Urine samples were collected separately during each of the three periods. At the end of the experiment, plasma samples were stored as described above and are shown in Appendix A. The animals were euthanized by injecting 5% KCl intravenously into the bolus.

### 4.5. RNS

To stimulate the renal nerve, we first localized and dissected this nerve, as described above. The renal nerve was placed on the bipolar electrode and then cut at the proximal part, so that the stimulation was only efferent. The electrode was connected to an electrical stimulator system (Grass S88,RI, USA), and the distal portion of the renal nerve was stimulated for 15, 1.5, and 0.5 ms.

### 4.6. Bic PVN Injection

Bic PVN microinjections were performed as described previously [34]. The rats were placed in a stereotaxic frame, and the PVN was located 1.8 mm caudal to the bregma, 0.5 mm lateral, and 7.8 mm deep (bite bar: −3.4 mm). Unilateral microinjections into the PVN were performed using glass micropipettes with tip diameters of 10–20 μm connected to a nitrogen pressure injector (MicroData Instruments, South Plainfield, NJ, USA). Microinjections were comprised of Bic methiodide (4 mM) in a volume of 100 nL. All drugs were dissolved in sterile saline.

### 4.7. Analysis of Baseline rSNA

Renal nerve activity was analyzed offline using a software (PowerLab, ADInstruments). The filtered nerve signal was rectified using only positive absolute values and then smoothed into a single line to obtain the mV/s. We assumed that the baseline signal represented 100%. We then compared the period in which the rSNA changed after Bic PVN. The background level of rSNA was determined by intravenous hexamethonium bromide administration (30 mg/Kg) as previously reported [14].

### 4.8. Plasma and Urine Analyses

At the end of the experiment, blood and urine samples and the kidneys were collected for subsequent analysis. A blood sample (approximately 1 mL) was collected from the femoral artery, and the left kidney was extracted. Blood samples were centrifuged at 3000× *g* for 20 min at 4 °C, and the plasma was stored at −80 °C. The kidneys were kept for up to 5 h in ice-cold PBS (150 mM NaCl, 2.8 mM sodium phosphate monobasic, 7.2 mM sodium phosphate dibasic at pH 7.4) containing protease (0.7 mg/mL pepstatin, 0.5 mg/mL leupeptin, and 40 mg/mL PMSF) and phosphatase (15 mM sodium fluoride and 50 mM sodium pyrophosphate) inhibitors.

The UFR was determined by measuring the urinary volumes normalized to the body weight obtained over a period. UNa concentrations were measured with an electrolyte analyzer (AVL Medical Instruments, Saint Quen l’Aumone, France) and then multiplied by the UFR to obtain the UNa. The CrCl was used to estimate glomerular filtration rates. Plasma and urinary creatinine concentrations were measured using a kinetic method (Labtest, Minas Gerais, Brazil) with a Thermo Plate Analyzer Plus (Thermo Fisher Scientific, Waltham, MA, USA).

### 4.9. Renal Cortical Membrane Protein Isolation

The kidneys were homogenized in ice-cold PBS. The kidney cortices were isolated at 4 °C and homogenized in PBS. The homogenate was centrifuged at 4000× *g* for 10 min at 4 °C, and the supernatant was removed and subjected to an additional 90 min of centrifugation at 28,000× *g* at 4 °C. The supernatant was discarded, and the pellets were resuspended in fresh PBS containing protease and phosphatase inhibitors. Protein concentrations were measured using the Lowry method [35].

### 4.10. Sodium Dodecyl Sulfate-Polyacrylamide Gel Electrophoresis and Immunoblotting

Renal cortical membrane proteins were solubilized in sample buffer (2% sodium dodecyl sulfate, 20% glycerol, 100 mM 2-mercaptoethanol, 50 mM Tris, pH 6.8, 0.05% bromophenol blue) and separated using sodium dodecyl sulfate-polyacrylamide gel electrophoresis on a 7.5% polyacrylamide gel. Proteins were then transferred from the polyacrylamide gel to a polyvinylidene difluoride microporous membrane (Immobilon, Millipore, Billerica, MA, USA). The membranes were blocked for 1 h with blocking solution (5% non-fat dry milk and 0.1% Tween 20 in PBS, pH 7.4) and incubated overnight with 1:1000 anti-NHE3, 1:1000 anti-PS552-NHE3 (phosphor specific anti-NHE3 monoclonal antibody directed against phospho-serine 552), or 1:5000 anti-actin. The membranes were washed five times with blocking solution and incubated for 1 h with the appropriate horseradish peroxidase-conjugated immunoglobulin secondary antibody (1:2.000). After five washes in blocking solution, the membranes were rinsed twice in PBS and incubated with enhanced chemiluminescence reagent for 1 min. The membranes were digitized on an ImageScanner LAS 4000 mini (GE Healthcare, Little Chalfont, UK) and quantified using ImageJ software (National Institutes of Health, Bethesda, MD, USA).

### 4.11. Statistics Analysis

All results are reported as the mean ± SE. Comparisons among groups were performed using one-way analysis of variance for multiple comparisons followed by Tukey’s test or Student’s *t*-test. A *p*-value < 0.05 was considered to indicate statistically significant results.

## 5. Conclusions

Our results suggest that an acute increase in BP has a more powerful influence on renal sodium and water absorption than the effect of acute stimulation of rSNA. Under both conditions, NHE3 phosphorylation regulation at serine 552 appears to play a critical role in sodium reabsorption.

## Figures and Tables

**Figure 1 ijms-24-00349-f001:**
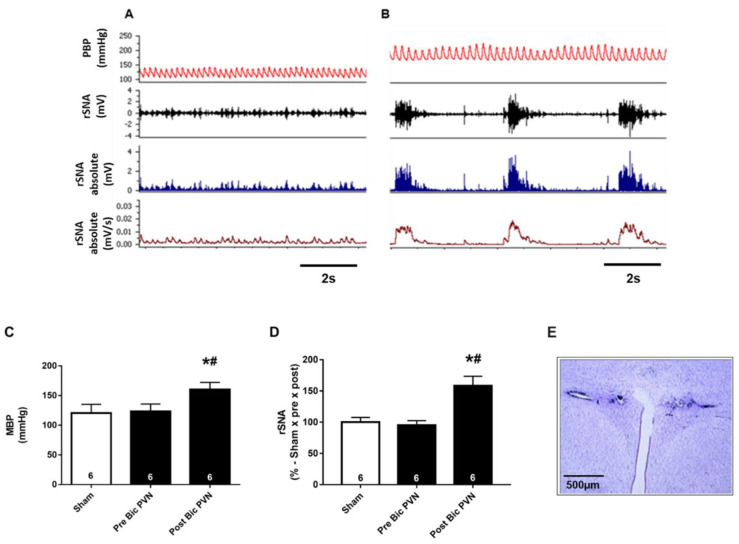
Effect of bicuculline injection into the paraventricular nucleus of the hypothalamus (Bic PVN) on mean arterial pressure (MAP) and renal sympathetic nerve activity (rSNA). (**A**) Typical trace containing pulsatile blood pressure (PBP) and rSNA of sham (whole experiment) and Bic PVN (at baseline before bicuculline infusion) groups. (**B**) PBP of the Bic PVN group after bicuculline infusion. Filtered signal from rSNA (black trace, second from top to bottom in (**A**,**B**) was first converted to positive absolute values (rSNA absolute, blue traces third from top) and then to mV/s (rSNA absolute mV/s, red lines bottom traces); see methods for more details. White bar represents the MAP of the sham group during with the experiment, and black bars represent the MAP of the Bic PVN group before and after Bic injection (**C**). White bar represents the MAP of the sham group during the experiment; black bars represent the rSNA of the Bic PVN group before and after Bic infusion (**D**). Representative histological image: arrow indicates the site of microinjection into the PVN. Scale bar = 500 μm. 3 V, third ventricle (**E**). The number of rats is indicated inside the bars. ** p* < 0.001 compared to sham. *# p <* 0.001 compared to pre-Bic PVN. One-way analysis of variance for multiple comparisons followed by Tukey’s test.

**Figure 2 ijms-24-00349-f002:**
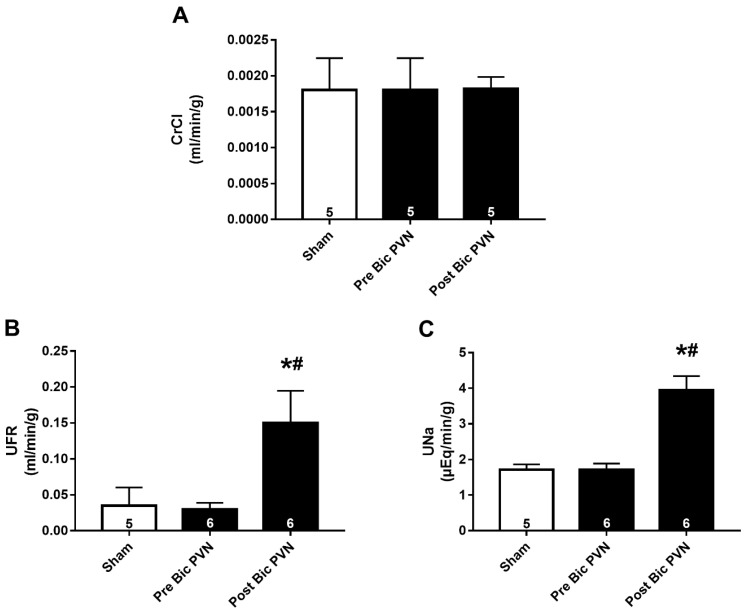
Effect of bicuculline on the paraventricular nucleus of the hypothalamus (Bic PVN) on renal function. (**A**) Creatinine clearance (CrCl) of rats in the sham, pre-Bic PVN, and post-Bic PVN groups. (**B**) White bar represents the urinary flow rate (UFR) of rats in the sham group during the experiment, and black bars represent the UFR of rats in the Bic PVN group before and after Bic injection. (**C**) White bar represents the urinary sodium rate (UNa) of rats in the sham group during the experiment, and black bars represent the UFR of rats in the Bic PVN group before and after Bic injection. The number of rats is shown inside the bars. * *p* < 0.001 compared to sham. # *p* < 0.001 compared to pre-Bic PVN. One-way analysis of variance for multiple comparisons followed by Tukey’s test.

**Figure 3 ijms-24-00349-f003:**
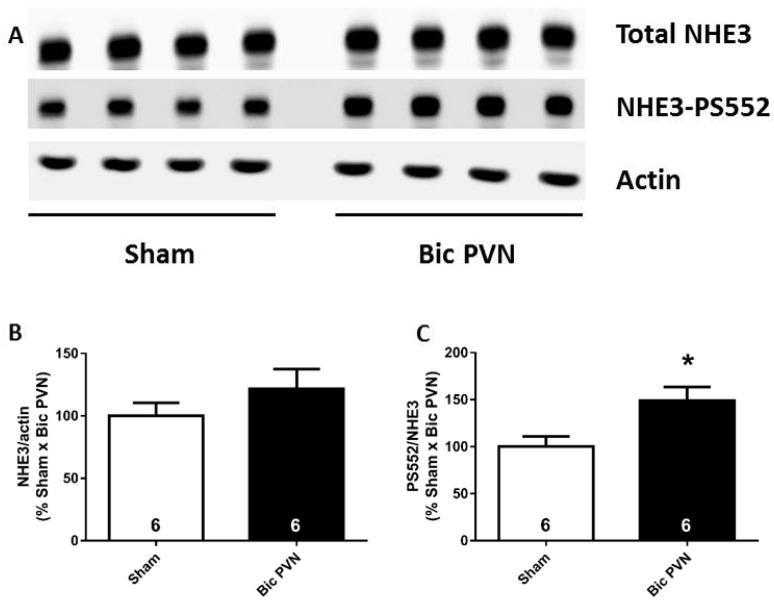
Effect of bicuculline on the paraventricular nucleus of the hypothalamus (Bic PVN) on sodium-hydrogen exchanger isoform 3 (NHE3). (**A**) Blots from the renal cortex of total NHE3, NHE3 phosphorylated at serine 552 (NHE3-PS552), and actin. The total amount of NHE3 corrected by actin did not distinguish between the sham and Bic PVN groups (**B**). Ratio of PS552/total NHE3 was significantly higher in the Bic PVN group (**C**). The number of rats is shown inside the bars. * *p* < 0.05 compared to sham. Student’s *t*-test.

**Figure 4 ijms-24-00349-f004:**
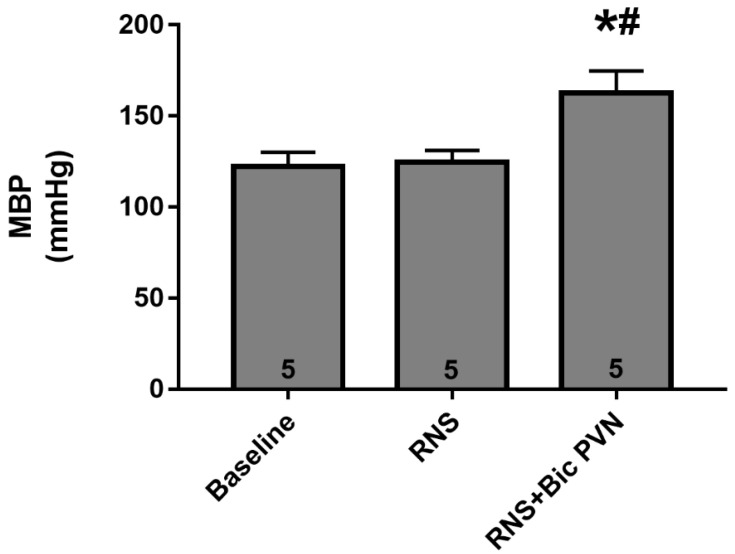
Mean arterial blood pressure (MBP) during the experiment at three different time points. Bars represent the mean arterial pressure (MAP) during the baseline period, under renal nerve stimulation (RNS), and under RNS plus bicuculline into the PVN (Bic PVN). The number of rats is shown inside the bars. * *p* < 0.001 compared to baseline. # *p* < 0.001 compared to RNS. One-way analysis of variance for multiple comparisons followed by Tukey’s test.

**Figure 5 ijms-24-00349-f005:**
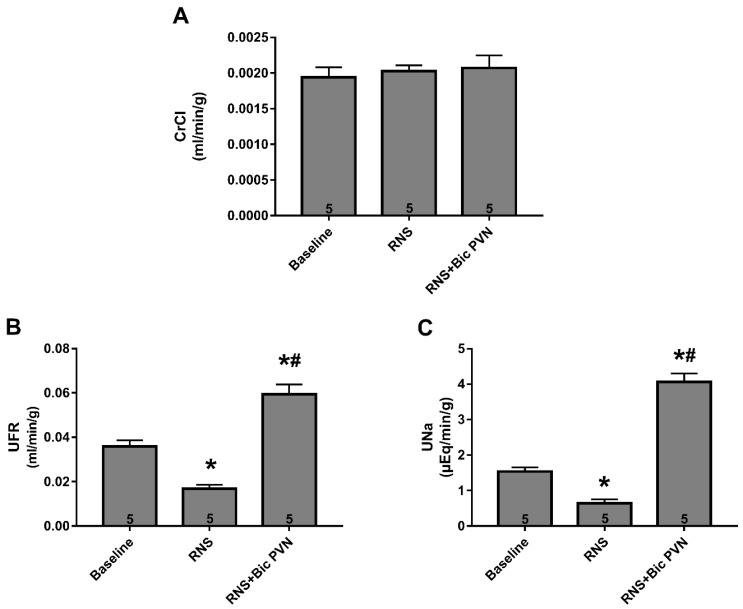
Creatinine clearance (CrCl), urinary flow rate (UFR), and urinary sodium rate (UNa) at three time points. (**A**) CrCl at baseline, under renal nerve stimulation (RNS), and under RNS plus bicuculline injection into the paraventricular nucleus of the hypothalamus (Bic PVN). (**B**) UFR at baseline, under RNS, and under RNS + Bic PVN. (**C**) UNa at baseline, under RNS, and under RNS + Bic PVN. The number of rats samples is shown inside the bars. * *p* < 0.001 compared to baseline. # *p* < 0.001 compared to RNS. One-way analysis of variance for multiple comparisons followed by Tukey’s test.

**Figure 6 ijms-24-00349-f006:**
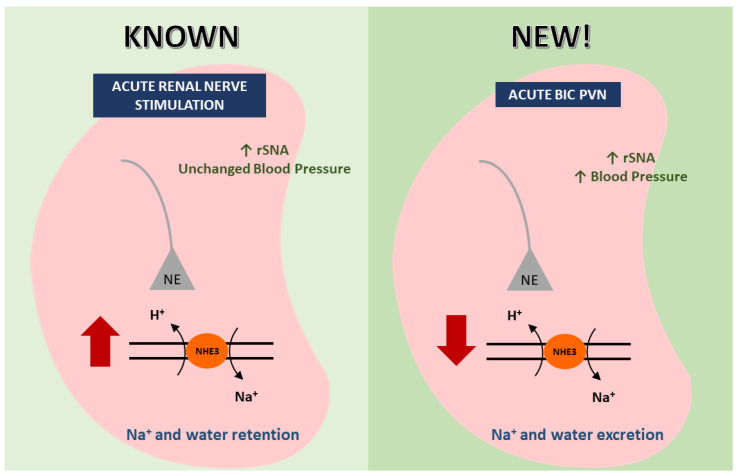
Schematic representation on the left of the figure shows increased NHE3 activity during selective renal nerve stimulation leading to increased sodium and water retention as previously reported [8]. On the right of the figure, shows that the renal nerve stimulation accompanied by increased blood pressure during PVN activation leads to reduction of NHE3 activity and consequently, increased salt and water excretion.

## Data Availability

Not applicable.

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
