# Peer review of "Relative Contribution of Blood Pressure and Renal Sympathetic Nerve Activity to Proximal Tubular Sodium Reabsorption via NHE3 Activity"

_ijms, 2022, doi:10.3390/ijms24010349_

Round 1

Reviewer 1 Report

1. Consider significant revision of the abstract. As written it is difficult to follow, all data and statistical significance is not included. Provide relevance and justification in the abstract. The role of targeting the PVN is not clear in the abstract as written.

2. General comment: please carefully go through the manuscript and correct all spelling and grammatical errors. Strengthen scientific writing style.

3. Limitations of the study and experimental design must be more clearly acknowledged in the discussion. The study was completed in anesthetized animals, would similar results be anticipated in conscious animals?

4. Sodium thiopental anesthesia has known effects on cardiovascular function and sympathetic nervous activity. Why was this anesthetic chosen? Dosage must be clear, especially where additional anesthetic is administered. How is this standardized between animals? This must be acknowledged in the discussion.

5. Activation of renal sympathetic nerves is known to increase activity of Na/K ATPase, how may this influence the interpretation of the study results?

6. Conceptual issue: in the context of hypertension, the homeostatic setpoint for blood pressure regulation is altered, such that a chronic elevation of blood pressure is sustained. If blood pressure is increased beyond this new setpoint, RSNA is suppressed and vice versa. The authors justify the rationale for the study by indicating that in models of hypertension and in hypertensive humans blood pressure and RSNA are both increased. However, the study results indicate that blood pressure elevation overrides the action of RSNA at the renal tubule. How can this observation be reconciled with the situation in these hypertensive models/humans? Consider elaborating on this point to clarify an explanation for the reader. This was rather unclear, particularly at the end of the introduction and discussion sections.

7. line 281 in methods, please provide a reference to support the statement indicating possible effects of the contralateral kidney (ie. U.C. Kopp, renorenal reflex)

8. How was RSNA verified? Was it entrained to the baroreflex? Postmortem recording? Please clarify.

9. In the results section, consider revising section headings to read as a descriptive statement which reflects the presented data. This is often more effective than a general statement such as "Effect of A on B"

Reviewer 2 Report

The study evaluated the effect of BP and renal sympathetic nerve activation in water and sodium handling by the kidney. Researchers performed experiments to assess the impact of BP and RNS activation separately and in the context of NHE3 activity alterations. The study is well-designed, includes a lot of work, and leads to interesting results.

I have the following comments.

1.     The authors should recheck the manuscript for language errors. The abstract should be rewritten in a way that would be easily understandable to non-experts. The references should be cited according to the journal guidelines, without “(see also…)”.

2.     A figure that depicts the findings of the manuscript would be well-come.

3.     The NHE3 results are impressive, but no specific inhibitor or activator of NHE3 was used to confirm that this exchanger is the only one responsible for the observed differences in diuresis and natriuresis. This limitation should be noted in the revised manuscript.

4.     Besides the significance of the results in understanding kidney physiology, do the authors believe these results could help the clinic in the near future?

Round 2

Reviewer 1 Report

Thank you for thoughtfully addressing my comments. The paper is much improved. Congratulations on your work.